# Application of a Stub-Loaded Square Ring Resonator for Wideband Bandpass Filter Design

**Ping Zhang [1], Liqin Liu [1], Deli Chen [1], Min-Hang Weng [1] and Ru-Yuan Yang [2],***

[1]    School of Information Engineering, Putian University, Putian 351100, China;
zhangpingdudu@gmail.com (P.Z.); llqsmile006@gmail.com (L.L.); 13860919709@139.com (D.C.);
hcwweng@gmail.com (M.-H.W.)
[2]    Graduate Institute of Materials Engineering, National Pingtung University of Science and Technology,
Pingtung County 912, Taiwan
*    Correspondence: ryyang@mail.npust.edu.tw; Tel.: +886-8-770-3202 (ext. 7555); Fax: +886-8-774-0552

**Abstract:** In this paper, a stub-loaded square ring resonator (SLSRR) is analyzed and applied to design a very simple and compact wideband bandpass filter structure. Resonant modes dependent on the structure parameters of the SLSRR are analyzed first, and then the first two modes are used to achieve a required passband. The input and output terminals are supplied with high impedance and strong coupling to provide sufficient coupling energy. Two wideband filter examples are designed, manufactured, and measured using the SLSRRs. The first filter is a wideband filter with a wide upper stopband, and the second filter is a dual wideband filter with a notched stopband between two passbands. The two filter examples are designed, fabricated, and measured to verify the design concept and present the advantages of easy design and a simple and compact structure.

**Keywords:** stub-loaded square ring resonator; resonant mode; bandpass filter; wideband

## 1. Introduction

With the increased development of wireless communication technology, research of the bandpass filter (BPF) has become an important development direction. At present, most of the research and applications of the BPF still focus on the broadband and multi-band design [1,2]. In the past, several resonator structures have been proposed to achieve the wideband filter design, such as the use of the step impedance resonator (SIR) [3,4], defect ground structure (DGS) [5], multi-layer structure [6], multi-mode (MMR) [7–9], stub-loaded resonator (SLR) [10–13], coupled-line [14,15], multiple-mode split-ring resonators (SRR) [16,17], and so on.

### 1.1. Related Work

In [3,4], the SIR was used to effectively obtain broadband performance by controlling harmonic frequencies. The typical structure of the SIR with two-step discontinuities causes larger radiation loss. Asymmetric SIR with only one-step discontinuity was thus proposed to achieve a wideband filter with low loss, wide stopband, and high selectivity. In [5], the defect ground structure (DGS) was used for wideband filter design, but the destruction of the ground plane would actually cause signal incompleteness. In [6], a wideband bandpass filter was reported using a multi-layer structure. The manufacturing procedure was complex. In [7–9], a multi-mode resonator (MMR) was used in wideband filters with band selectivity, but still exhibited large circuit sizes and insertion loss. In [10–13], wideband BPFs were designed using SLRs. In [10], a short-circuited microstrip stub-loaded ring resonator (SC-SLRR) with a taped line was used to form a wide frequency band. In [11], a single wavelength ring SIR was introduced to have a wide frequency band. In [12], a wideband BPF with an

improved upper stopband was realized using open/shorted stubs. In [13], a new type of open-circuit stub was applied to the square ring resonator to form a dual-frequency and triple-frequency filter. In [14], a pair of open-circuited stubs and three pairs of coupled-line sections were used to produce a UWB BPF. In [15], a UWB BPF with a notched band and a wide upper stopband was realized by using two pairs of half-wavelength, high impedance line resonators. However, the bandwidth is controlled by the coupling gap and thus large bandwidth would be limited due to the requirement of the precious fabricated technology. In [16,17], wideband BPFs were implemented using multiple-mode split-ring resonators in a rectangular waveguide cavity. However, the above design procedure and/or structure are complex. Therefore, designing a wideband filter without using a complicated design process and structure while achieving good band performance is desired. In the past, the stub-loaded square ring resonator (SLSRR) was used to design a dual-band BPF [18]. The filter structure is very simple and comprises only a stub-loaded square ring resonator with two stubs.

### 1.2. Contributions

In this paper, a stub-loaded square ring resonator is discussed in detail and then applied to design wideband bandpass filters with a very simple and compact structure. The high impedance input and output are used to provide a strong coupling energy to the resonator. In order to verify the convenience and effectiveness of this SLSRR structure for filter design, two wideband filter examples using the SLSRRs are designed in this study. The first filter is a wideband filter with a wide upper stopband, and the second filter is a dual wideband filter with a notched stopband between two passbands. This design uses a low-cost commercial FR4 substrate, having a thickness of 0.8 mm, dielectric constant ($\varepsilon$r) of 4.4, and loss tangent (tag$\delta$) of 0.02. A full-wave electromagnetic (EM) software tool IE3D is used to simulate the responses of the designed filter [19]. The two filter examples are designed, fabricated, and measured to verify the design concept. The method for choosing two resonant modes to be the passband edges of the wideband response is described clearly in this paper. The two presented filter examples both show the advantages of easy design and simple and compact structure. The filters are fabricated and measured to verify the design concept. The measured results generally have a good agreement with the simulated results.

### 1.3. Organization of the Paper

The organization of this paper is shown as follows. In Section 1, the background, related works, and novelty of this study are described. In Section 2, the resonant modes of the odd and even equivalent circuits of the SLSRR are analyzed in detail. By setting the electrical length ratio, the resonant modes of the SLSRR can be controlled depending on the added stub and tuned effectively to the desired frequency. In Section 3, the filter design method using the SLSRR for a wideband filter with a wide stopband and a dual wideband filter with a notched stopband is addressed. These filters are fabricated and measured individually. A table is also presented to compare the proposed filters to several reported wideband filters. In Section 4, a conclusion is made to show the finding and benefits of the proposed design.

### 2. Analysis of Resonant Mode

Figure 1 shows the structure of the stub-loaded square ring resonator. The square ring resonator has the physical lengths ($2L_2 + 4L_3$) and the width ($W_2$) and two stubs with physical length ($L_1$) and the width ($W_1$). Namely, the impedances of two stubs and the square ring resonator can be set differently and expressed as $Z_1$ and $Z_2$, respectively. Because the used resonator is uniformly symmetrical with the S-S' symmetry line, as shown in Figure 1, an electric wall (EW) or a magnetic wall (MW) appeared in odd mode or a uniform mode excitation, respectively, in the symmetrical plane [20]. Thus, its resonant property can be analyzed by even mode and odd mode analysis. Figure 2 displays the odd/even mode equivalent circuits of the square ring resonator loaded with two stubs. The physical lengths ($L_1$, $L_2$, and $L_3$), also expressed as the electrical lengths ($\theta_1$, $\theta_2$, $\theta_3$), are the varied parameters of the $Y_{inodd}$

and $Y_{ineven}$. Thus, by using the transmission line theory [20], the input admittance of the odd mode ($Y_{inodd}$) and even mode ($Y_{ineven}$) can be driven, which are shown in Equations (1) and (2) [18].

$$Y_{inodd} = j\frac{tan\theta_1 - cot\theta_3 - \frac{Z_1}{Z_2}\cot(\theta_2 + \theta_3)}{Z_1\left[1 + tan\theta_1 cot\theta_3 + \frac{Z_1}{Z_2}tan\theta_1 \cot(\theta_2 + \theta_3)\right]} \tag{1}$$

$$Y_{ineven} = j\frac{tan\theta_1 + tan\theta_3 + \frac{Z_1}{Z_2}\tan(\theta_2 + \theta_3)}{Z_1\left[1 - tan\theta_1 tan\theta_3 - \frac{Z_1}{Z_2}tan\theta_1 \tan(\theta_2 + \theta_3)\right]} \tag{2}$$

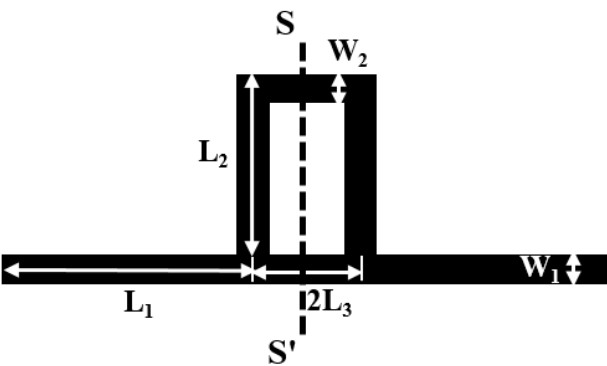

**Figure 1.** The structure of the stub-loaded square ring resonator (SLSRR).

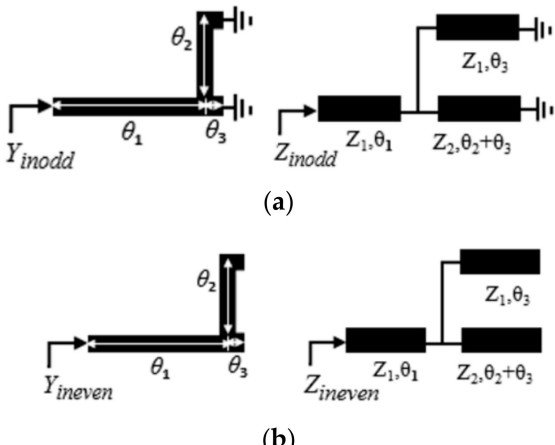

(a)

(b)

**Figure 2.** (**a**) Odd mode and (**b**) even mode equivalent circuits of the SLSRR.

To accurately analyze the odd–even mode, Im [$Y_{inodd}$] and Im [$Y_{ineven}$] are both set as 0 to be the resonant conditions. Thus, the following relationship for the odd and even mode are expressed as Equations (3) and (4), respectively:

$$tan\theta_1 - cot\theta_3 - \frac{Z_1}{Z_2}\cot(\theta_2 + \theta_3) = 0 \tag{3}$$

$$tan\theta_1 + tan\theta_3 + \frac{Z_1}{Z_2}\tan(\theta_2 + \theta_3) = 0 \tag{4}$$

In order to co-relate the electrical lengths ($\theta_1$, $\theta_2$, $\theta_3$), two new parameters are defined as followed:

$$\alpha_1 = \frac{2\theta_1}{2(\theta_1 + \theta_3)} = \frac{2\theta_1}{\theta_t} \tag{5}$$

$$\alpha_2 = \frac{2\theta_2}{2(\theta_1 + \theta_3)} = \frac{2\theta_2}{\theta_t} \tag{6}$$

Then, Equations (5) and (6) can be expressed as followed:
Odd mode:

$$\tan\left(\frac{\alpha_1 \cdot \theta_t}{2}\right) - \cot\left(\frac{(1-\alpha_1) \cdot \theta_t}{2}\right) - \frac{Z_1}{Z_2} \cot\left(\frac{(1-\alpha_1+\alpha_2) \cdot \theta_t}{2}\right) = 0 \tag{7}$$

Even mode:

$$tan\left(\frac{\alpha_1 \cdot \theta_t}{2}\right) + tan\left(\frac{(1-\alpha_1) \cdot \theta_t}{2}\right) + \frac{Z_1}{Z_2} tan\left(\frac{(1-\alpha_1+\alpha_2) \cdot \theta_t}{2}\right) = 0 \tag{8}$$

In this study, $Z_1 = Z_2$ is set to simply the design. Figure 3 shows the electrical length $\theta_t$ of the SLSRR as functions of $\alpha_2 = 0.25, 0.35, 0.45$ and $\alpha_1$ in the interval of 0, obtained by solving Equations (7) and (8) with the MATLAB tool. As shown in Figure 3, when $\alpha_1$ is increased, the first odd mode $f_{o1}$ and the first even mode $f_{e1}$ are gradually separated. Thus, if the coupling energy is sufficiently large, a wider passband can be formed by selecting the desired first odd mode $f_{o1}$ and the even mode $f_{e1}$. Thus, two wideband filter examples would be designed by using the SLSRRs. The first filter example is a wideband filter with a wide upper stopband, and the second filter example is a dual wideband filter with a notched stopband between two passbands. As using the resonator to design the filter, the physical lengths ($L_1$, $L_2$, and $L_3$) shall be determined first and used as the initial values to simulate for the required operating frequency.

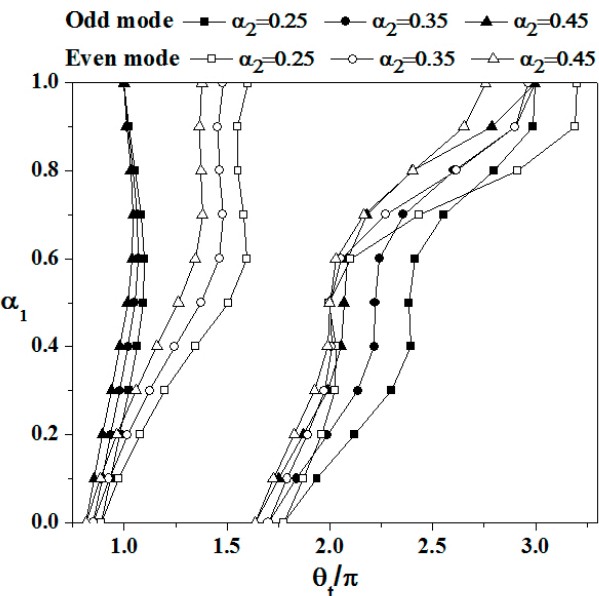

**Figure 3.** The electrical length $\theta_t$ of the SLSRR as functions of $\alpha_2 = 0.25, 0.35, 0.45$.

## 3. Filter Design

### 3.1. A Wideband Filter with a Wide Stopband

Figure 4 shows the structure of the wideband BPF with a wide upper stopband using only one SLSRR. This BPF basically consists of only one SLSRR and two ports of input (Port 1) and output (Port 2) with coupling length at the two sides. The first filter sets the center frequency of the passband around 4.0 GHz, and the 3 dB fractional bandwidth (FBW) larger 50%. Then, $f_{o1} = 3.66$ GHz and $f_{e1} = 5.34$ GHz are chosen at the passband edges in this design. As mapping to Figure 3, $\alpha_1 = 0.9$ and $\alpha_2 = 0.35$ can be initially selected. It is also found that the second odd mode $f_{o2}$ and the second

even mode $f_{e2}$ are shifted to the higher frequencies and far away from the first odd mode $f_{o1}$ and the first even mode $f_{e1}$, thus a wide upper stopband is expected as the passband is formed. The first odd mode $f_{o1}$ corresponds to $\theta_t = \pi$, and the first even mode $f_{e1}$ corresponds to $\theta_t = 1.46\pi$. By taking $Z_1 = Z_2 = 100\ \Omega$, the electrical lengths of the used SLSRR are determined as $\theta_1 = 100°$, $\theta_2 = 41.4°$, and $\theta_3 = 11.1°$; the corresponding physical lengths $L_1 = 10.6$ mm, $L_2 = 4.4$ mm, $L_3 = 1.2$ mm, $W_1 = 0.35$ mm, and $W_2 = 0.35$ mm are selected as the initial length values, respectively. The simulation works are processed by the full-wave EM software tool IE3D [19].

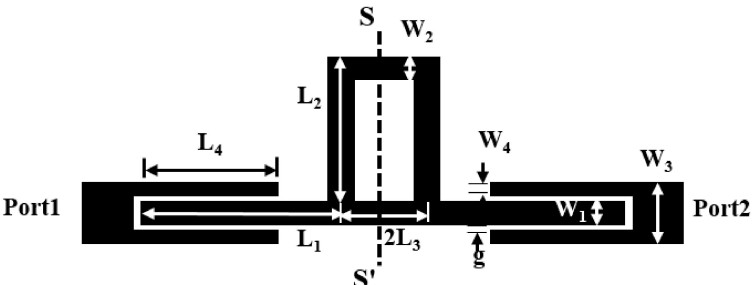

**Figure 4.** The structure of the first wideband bandpass filter (BPF) with a wide stopband using SLSRR.

Figure 5 shows the effect of different lengths $L_1$ on the frequency response of the SLSRR using EM simulation with weak coupling. The simulated result indicated the frequency of the resonant modes is controlled by adjusting $L_1$. The first two resonant modes are higher than other higher frequency modes. When the stub length $L_1$ is extended from 8.6 mm to 12.6 mm, the frequency of the first resonant mode is lowered from 5.2 GH to 3.9 GHz, and the frequency of the third mode is reduced from 12 GHz to 9.5 GHz. Moreover, with increasing the $L_1$, the first two resonant modes shift to a lower frequency at the same time and the frequency difference between the first two resonant modes is similar, indicating the bandwidth of the passband formed by the first two resonant modes is hard to tune.

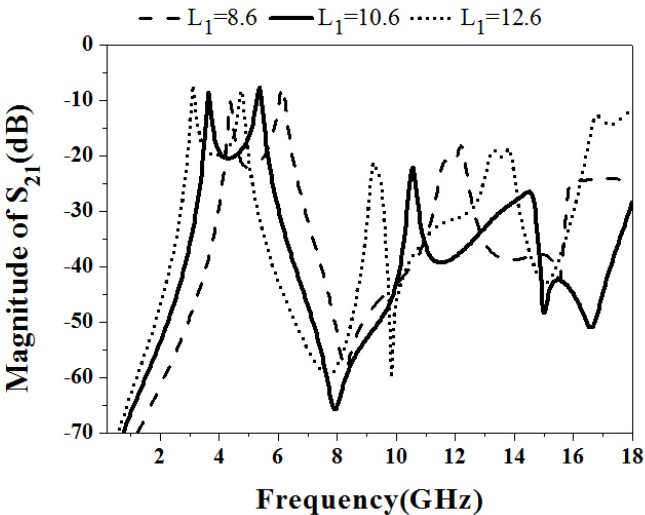

**Figure 5.** Effect of different length $L_1$ on frequency response of the SLSRR ($L_2 = 4.4$ mm, $L_3 = 1.2$ mm are fixed in this case).

Figure 6 displays the frequency response of the SLSRR with different ring lengths $L_2$ using EM simulation with weak coupling. It is found that by adjusting the length $L_2$, the position of the second resonant modes can be finely adjusted, but the first resonant mode is not affected. With decreasing $L_2$, $f_{o1}$ is fixed at 3.8 GHz, and $f_{e1}$ is shifted from 5.2 GHz to 6.2 GHz, while other higher resonant modes are also shifted to a higher frequency. Therefore, it is possible to tune the bandwidth of the passband formed by the first two resonant modes ($f_{o1}$ and $f_{e1}$).

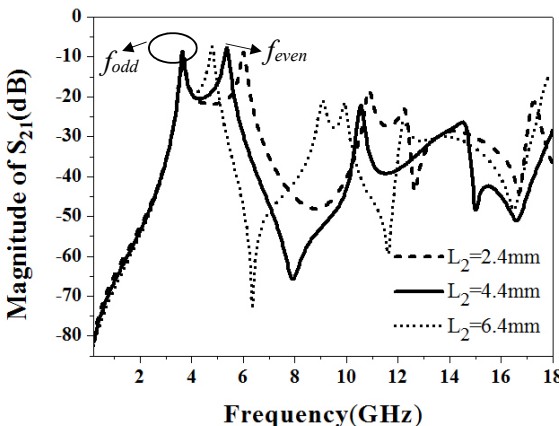

**Figure 6.** Effect of different ring length $L_2$ on the frequency response of the SLSRR ($L_1$ = 10.6 mm and $L_3$ = 1.2 mm are fixed in this case).

Figure 7 displays the frequency response of the SLSRR with different ring lengths $L_3$ using EM simulation with weak coupling in (a) wide frequency range and (b) narrow frequency range for two main resonant modes. It is found that $L_3$ is reduced, fo1 is slightly reduced, fe1 is slightly increased, and the third mode is shifted to the higher frequency. Therefore, the bandwidth of the passband can be sufficiently tuned by adjusting the length $L_3$.

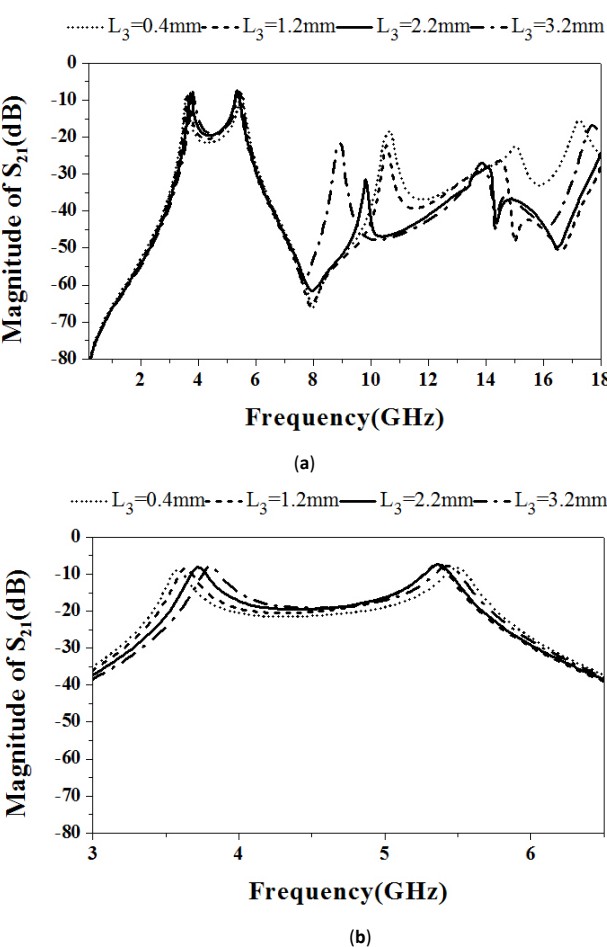

(a)

(b)

**Figure 7.** Effect of different ring length $L_3$ on the frequency response of the SLSRR in (**a**) wide frequency range and (**b**) narrow frequency range for two main resonant modes. (in this case, $L_1$ = 10.6 mm and $L_2$ = 4.4 mm are fixed).

Based on the design curves above, the frequencies of the first two resonant modes ($f_{o1}$ = 3.66 GHz and $f_{e1}$ = 5.34 GHz) can be selected at the passband edges to expect a wideband response. In order to obtain the response characteristics of the wideband bandpass filter, the input/output (I/O) terminal is fed in a bilateral strong coupling manner. The coupling length $L_4$ and the coupling gap (g) are adjusted to select an appropriate value.

Figure 8 shows the frequency responses of (a) $S_{21}$ and (b) $S_{11}$ of the proposed wideband BPF with various coupling lengths $L_4$. When the coupling length $L_4$ = 8.6 mm, the return loss in the filter passband is larger than 15 dB, the insertion loss is optimal as 0.5 dB, and the stopband can be extended from 7 GHz to 12 GHz.

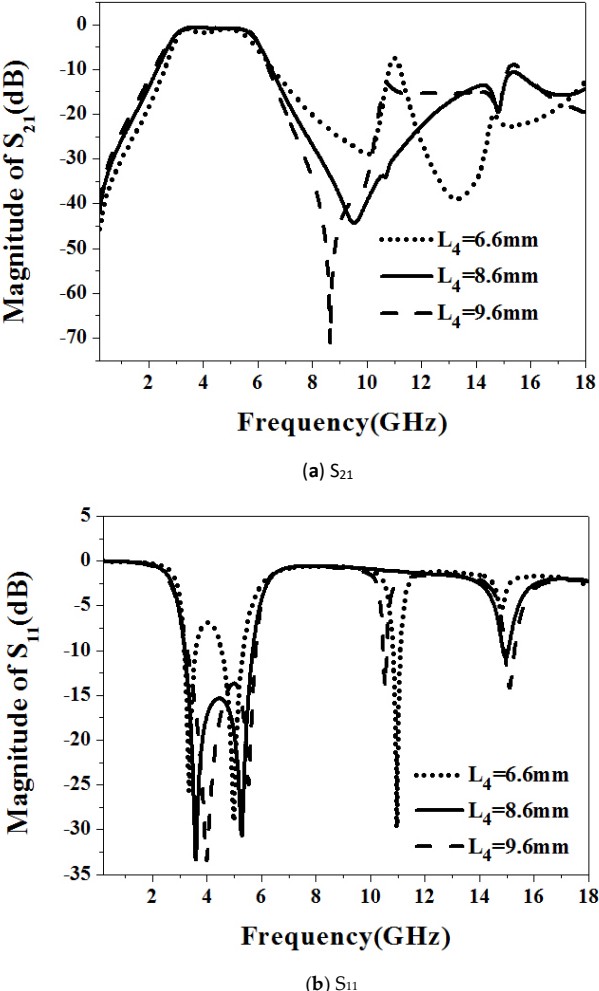

**Figure 8.** Wideband filter response of (**a**) $S_{21}$ and (**b**) $S_{11}$ with different coupling lengths $L_4$ (in this case, $L_1$ = 10.6 mm, $L_2$ = 4.4 mm, and $L_3$ = 1.2 mm).

Figure 9 shows the wideband response characteristics at different coupling gaps (g). It is clearly observed that the smaller the value of g, the stronger the coupling energy as well as the smaller the insertion loss in the passband. As the coupling gap ($g_1$) decreases from 1 mm to 0.15 mm, the passband is formed and the bandwidth is 60%. Considering the minimum engraving distance of the engraving machine is 0.15 mm, g = 0.2 mm is selected to facilitate the production. The designed component can achieve a center frequency of 4.38 GHz, a passband of 2.82 GHz to 5.97 GHz, and a stopband greater than 10 dB between 6.5 and 12 GHz. The upper stopband of this filter example provides an advantage of inserting another filter response in this stopband.

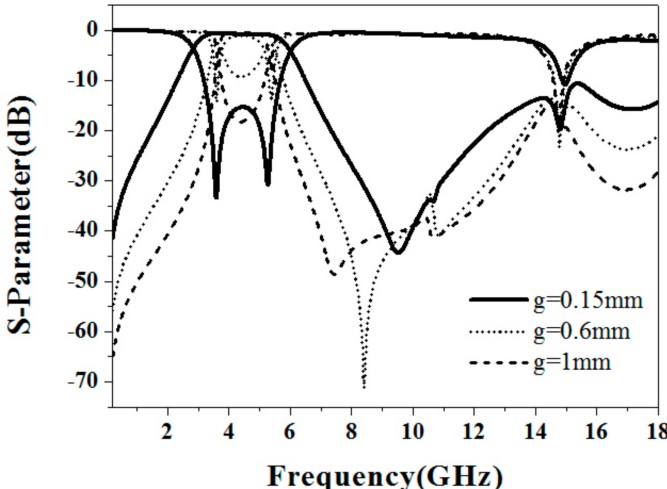

**Figure 9.** Wideband filter response characteristics for different coupling gaps (g) (in this case, $L_1$ = 10.6 mm, $L_2$ = 4.4 mm, $L_3$ = 1.2 mm, and $L_4$ = 8.6 mm).

In order to confirm the design concept, the designed filter was manufactured and measured. The actual fabrication dimensions are: $L_1$ = 10.6 mm, $L_2$ = 4.4 mm, $L_3$ = 1.2 mm, $L_4$ = 8.6 mm, $W_1$ = 0.35 mm, $W_2$ = 0.35 mm, $W_3$ = 1.53 mm, $W_4$ = 0.2 mm, g = 0.2 mm. The picture of the fabricated wideband BPF is as shown in Figure 10a. The overall circuit size is 12 mm × 4 mm, which is 0.28 λg × 0.09 λg, where λg is the guided wavelength of the center frequency.

The wideband frequency response was tested using the HP8722ES network analyzer. As shown in Figure 10b, the measured results include a fundamental frequency of 4.5 GHz, an average insertion loss of 1.6 dB, an average return loss of more than 12 dB, a 3 dB fractional bandwidth (FBW) of 60% (from 3.4 GHz to 6.4 GHz), and a stopband from 7 GHz to 12 GHz with a rejection greater than 20 dB. The maximum group delay variation in the passband is 0.4 ns, as shown in Figure 10c.

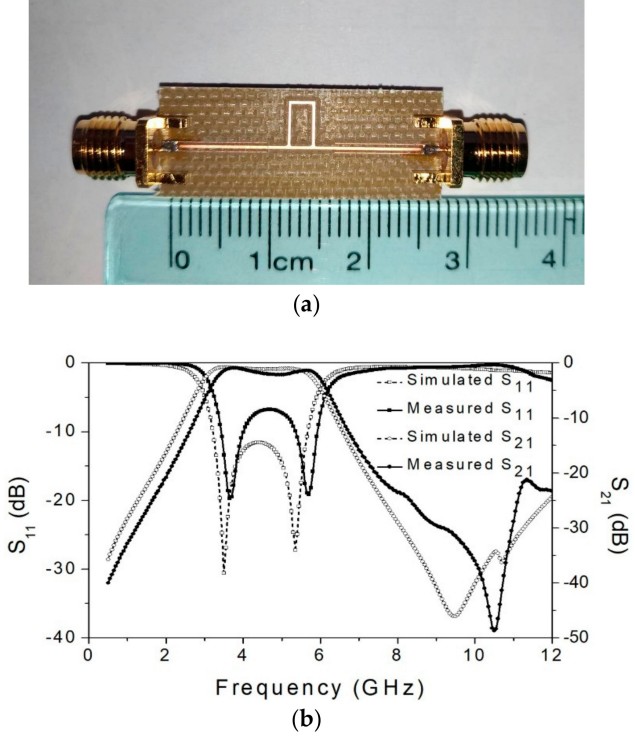

**Figure 10.** *Cont*

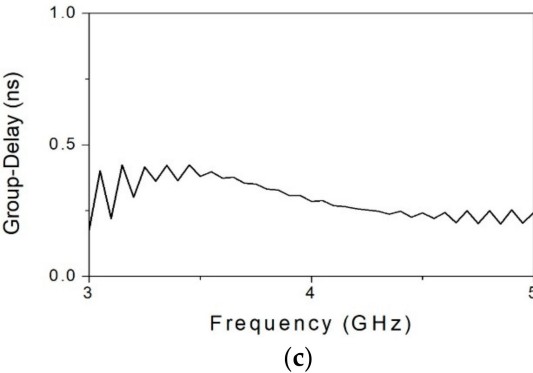

**Figure 10.** The first wideband BPF example. (**a**) Photograph, (**b**) simulated and measured results, and (**c**) group delay.

### 3.2. A Dual Wideband Filter with a Notched Stopband

Figure 11 shows the structure of the w second dual wideband BPF using one SLSRR. This BPF basically consists of two SLSRRs (SLSRR 1 and SLSRR 2) and two ports of input (Port 1) and output (Port 2) with coupling lengths at the two sides. The second filter example has dual wideband responses, which the first and second responses resulted from SLSRR 1 and SLSRR 2 are set at 4 and 8.0 GHz, and 10 dB and 3 dB BW of 52% and 56%, respectively.

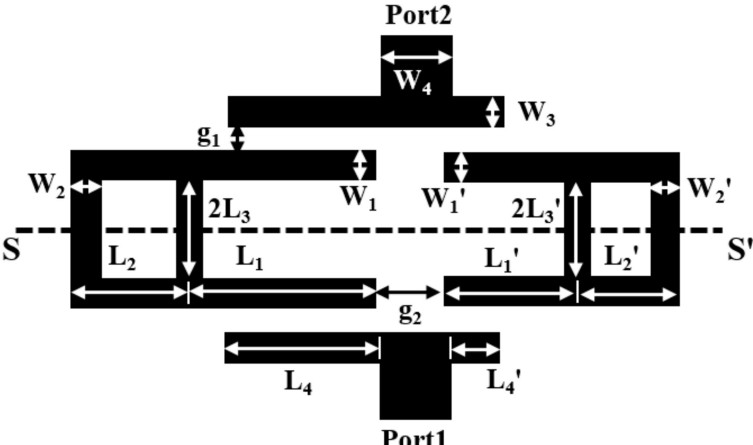

**Figure 11.** The structure of the second dual wideband BPF using two SLSRRs (SLSRR 1 is on the left for the first wideband and SLSRR 2 is on the right for the second wideband).

In the first wideband response, an upper stopband is required since the second wideband response would be set in the stopband. Based on the design addressed in Section 3.1, the first wideband response is set to have a center frequency at 4 and with a 3 dB BW of 50%. Thus, the frequencies of the first two resonant modes ($f_{o1}$ = 3.4 GHz and $f_{e1}$ = 4.6 GHz) of the SLSRR can be selected at the first wide passband edges to expect a wideband response. Based on the design curves of Figure 3, $\alpha_1$ = 0.92 and $\alpha_2$ = 0.35 can be initially selected. By taking $Z_1$ = $Z_2$ = 100 Ω, the electrical lengths of the used SLSRR 1 are determined as $\theta_1$ = 102°, $\theta_2$ = 40.2°, and $\theta_3$ = 5.9°; the corresponding physical lengths $L_1$ = 12.2 mm, $L_2$ = 4.8 mm, $L_3$ = 0.7 mm, $W_1$ = 0.35 mm, and $W_2$ = 0.35 are selected as the initial length values, respectively. The design steps of the first wideband are similar to those described in Section 3.1. The following steps are addressed for the second wideband.

For the second wideband response, the center frequency is set at 8.0 with a 3 dB BW of 56%, thus the frequencies of the first two resonant modes ($f_{o1}$ = 6.5 GHz and $f_{e1}$ = 9.1 GHz) can be selected for at the first wide passband edges to expect a wideband response. Based on the design curves of Figure 3, $\alpha_1$ = 0.9 and $\alpha_2$ = 0.25 can be initially selected. Similarly, by taking $Z'_1$ = $Z'_2$ = 100 Ω, the

electrical lengths of the used SLSRR 2 are determined as $\theta_1 = 103.3°$, $\theta_2 = 31.2°$, and $\theta_3 = 11.5°$; the corresponding physical length $L_1 = 6.3$ mm, $L_2 = 1.9$ mm, $L_3 = 0.7$ mm, $W_1 = 0.35$ mm, and $W_2 = 0.35$ are selected as the initial length values, respectively.

　　　Figure 12 shows the effect of different length $L_{1'}$ on the frequency response of the SLSRR 2 using EM simulation with weak coupling. When $L_1 = 6.3$ mm, the center frequency of the passband is 8 GHz. Figure 13 shows the effect of different ring lengths $L_{2'}$ on the frequency response of the SLSRR 2 using EM simulation with weak coupling. It is found that as $L_2$ decreases, $f_{odd}$ is fixed at 6.5 GHz, and $f_{even}$ shifts from 8.2 GHz to 10.2 GHz, which would increase the passband bandwidth in the filter response. Thus, an $L_2$ of 1.9 mm is suitable to adjust the bandwidth between 6 and 10 GHz.

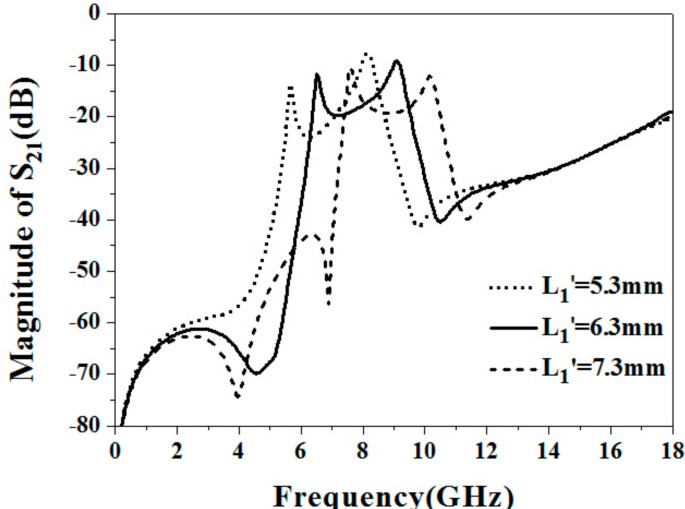

**Figure 12.** Effect of different length $L_{1'}$ on frequency response of the SLSRR 2 ($L_{2'} = 1.9$ mm, $L_{3'} = 0.7$ mm are fixed in this case).

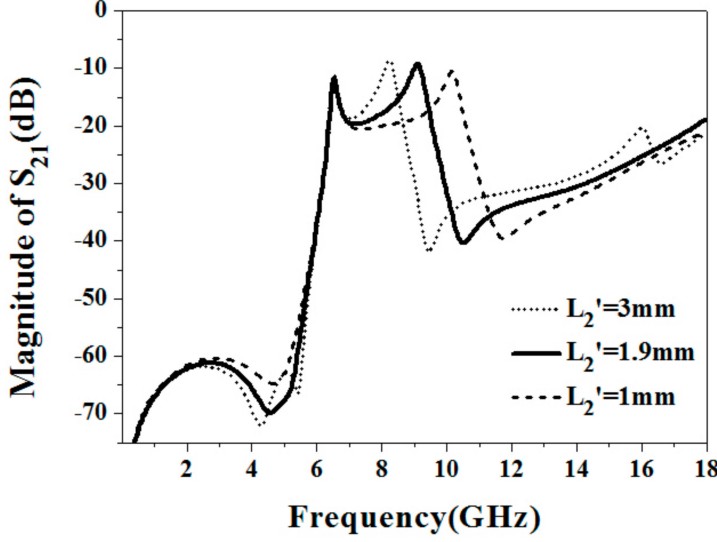

**Figure 13.** Effect of different ring length $L_{2'}$ on the frequency response of the SLSRR 2 ($L_{1'} = 6.3$ mm and $L_{3'} = 0.7$ mm are fixed in this case).

　　　Figure 14 shows the combination of the simulated performance of individual ultra-wideband responses from 3 to 5 GHz and 6 to 10 GHz, respectively. For dual wideband filter design, two coupled lines connected to Port 1 and Port 2 are simultaneously coupled to the SLSRR 1 and SLSRR 2, which achieve not only the strong coupling but also extra transmission zeros near the passband edges.

Coupling gap ($g_1$) and coupling lengths are important parameters for providing strong coupling. $L_4$ and $L_{4'}$ are adjusted to have a good impedance match between the SLSRR 1 and SLSRR 2. Therefore, to achieve the required dual wideband target, the coupling gap and coupling lengths can be obtained as $g_1 = 0.2$ mm, $g_2 = 0.8$ mm, $L_1 = 11.6$ mm, $L_2 = 3$ mm, $L_3 = L_{3'} = 1$ mm, $L_{1'} = 5.7$ mm, $L_{2'} = 1.9$ mm, and $L_4 = 11.5$ mm, $L_{4'} = 5$ mm, W1 = W2 = W3 = 0.35 mm, W3 = 0.2 mm, W4 = 1.53 mm.

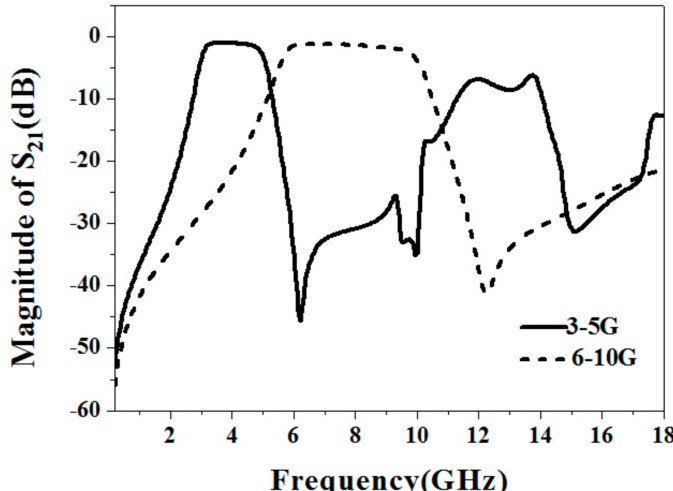

**Figure 14.** The combination of the simulated performance of individual ultra-wideband responses from 3 to 5 GHz and 6 to 10 GHz, respectively.

Figure 15 shows the simulated performance of dual wideband responses of the second filter example, using EM simulator tool IE3D. It is also found that when the initial size parameters of the SLSRR 1 and SLSRR 2 with two coupled lines are selected above, without spending any bandstop structure, a notched band at 5.15 GHz can be obtained, thus avoiding the interference of the signals in the wireless local area network (WLAN) system.

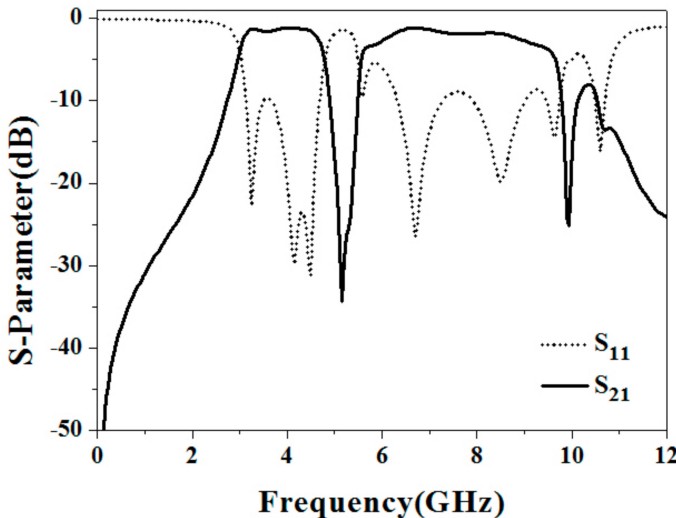

**Figure 15.** Simulated performance of dual wideband responses of the second filter example.

Figure 16 shows (a) photograph, (b) simulated and measured results, and (c) group delay of the second wideband BPF example. The second filter reaches two wideband responses in which the first and second responses are centered at 4.1 and 8.0 GHz, with an average insertion loss of more than

1.2 and 2.8 dB, average return loss of more than 15 dB and 10 dB, a 3 dB FBW of 46% and 55%, and maximum group delay variation in the passband of 0.5 nS and 0.3 ns, respectively.

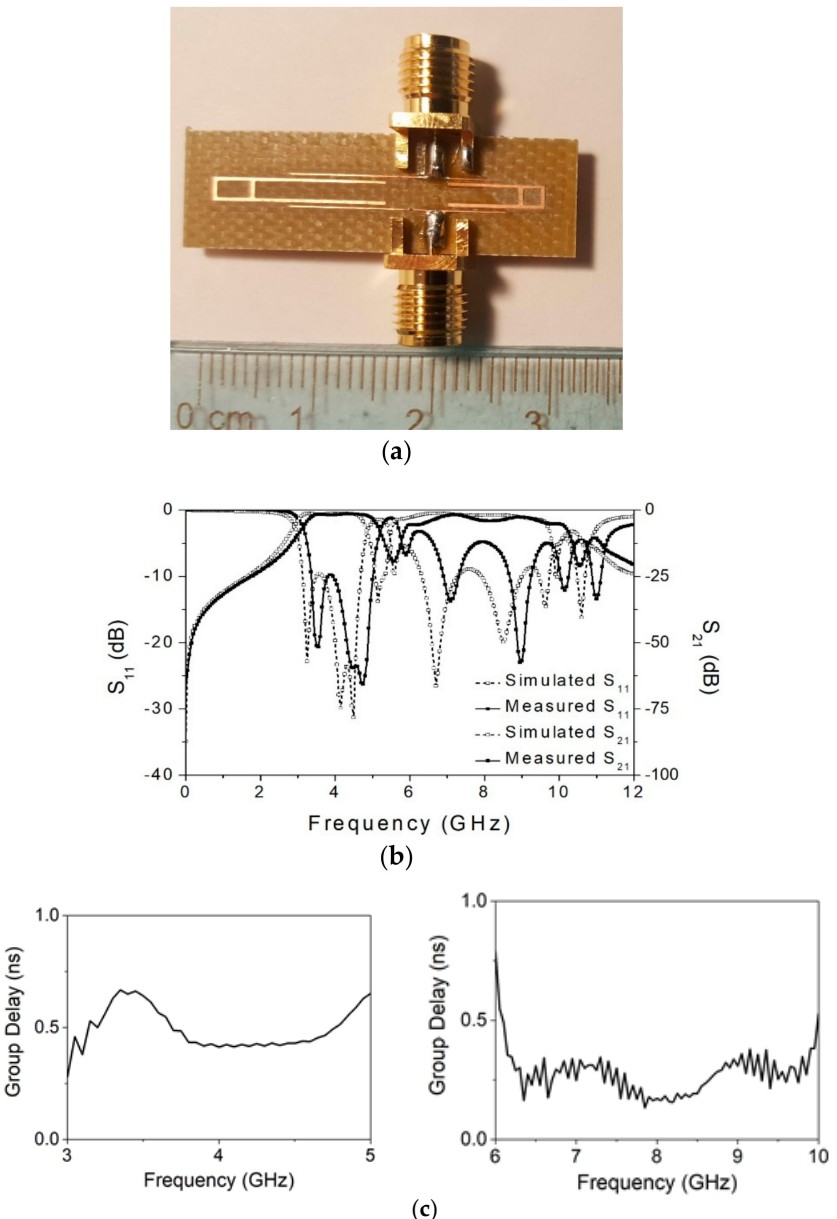

**Figure 16.** The second wideband BPF example. (**a**) Photograph, (**b**) simulated and measured results, and (**c**) group delay.

For both filter examples, the insertion loss and return loss are both satisfactory from the measurement data. Although the simulation results and the measured results have a slight mismatch, they still have an agreement. The main reason for the mismatch is that the needle of the engraving machine in this study is relatively thick, resulting in a large manufacturing error.

Table 1 compares the proposed filter to some previously reported wideband BPFs. The designed filter has acceptable filter responses as compared to other filters. Additionally, as compared to the previous works with a similar performance [15], Example 2 shows a miniaturized size and a simple configuration, without needing a defected ground area to destroy the ground plane.

**Table 1.** Comparison of filter performances of the proposed filter with the previous published works.

| | Ref. [5] | Ref. [6] | Ref. [11] | Ref. [12] | Ref. [14] | Ref. [15] | Example 1 | Example 2 |
|---|---|---|---|---|---|---|---|---|
| Center frequency (GHz) | 5 | 3 | 2.3 | 1 | 1 | 4/8 | 4.5 | 4.1/8.0 |
| $|S_{11}|$ (dB) | 12 | 11.7 | >13 | 15 | 15 | 10 | 10 | 10 |
| $|S_{21}|$ (dB) | 1.2 | 2.1 | 0.35 | 1 | 1 | 1/2 | 1.6 | 1.2/2.8 |
| 3 dB FBW (%) | 100 | 107 | 80 | 123 | 123 | 42/58 | 60 | 46/55 |
| Circuit Size ($\lambda g \times \lambda g$) | Unknown | 0.89 × 0.46 | 0.53 × 0.43 | 0.17 × 0.14 | 0.17 × 0.14 | 0.70 × 0.28 | 0.28 × 0.09 | 0.31 × 0.03 |
| Number of notched bands | 1 | 1 | 2 | 0 | 0 | 1 | 0 | 1 |
| Wide stopband | No | No | No | Yes | No | No | Yes | No |
| Defected ground | Yes | Yes | No | No | No | No | No | No |

## 4. Conclusions

In this paper, we present a simple way and structure to design different wideband bandpass filters. The main structure used in the filter is the stub-loaded square ring resonator with two stubs, which have much design freedom. The analysis of the odd–even resonant mode of the resonator dependent on the structure parameters of the resonator is discussed first and the first two resonant modes are used to form the passband edges of the wideband response. With the desired coupling scheme of the input and output terminals, sufficient coupling energy is provided. The first filter achieves a fundamental frequency of 4.5 GHz with an average insertion loss of 1.6 dB, an average return loss of more than 12 dB, a 3 dB fractional bandwidth (FBW) of 60% (from 3.4 GHz to 6.4 GHz), a maximum group delay variation in the passband of 0.4 nS, and a stopband rejection greater than 15 dB from 7 GHz to 12 GHz. The second filter reaches two wideband responses in which the first and second responses are centered at 4.1 and 8.0 GHz, with an average insertion loss of more than 1.2 and 2.8 dB, average return loss of more than 15 dB and 10 dB, a 3 dB BW of 46% and 55%, and maximum group delay variation in the passband of 0.5 ns and 0.3 ns, respectively. The simulation and the measurement show an agreement, verifying the design concept.

**Author Contributions:** Conceptualization, M.-H.W.; Methodology, P.Z. and D.C.; Software, P.Z.; Validation, P.Z. and L.L.; Formal analysis, P.Z and D.C.; Investigation, D.C. and M.-H.W.; Resources, R.-Y.Y.; Data curation, P.Z.; Writing—Original draft preparation, P.Z. and L.L.; Writing—Review and editing, P.Z. and R.-Y.Y.; Visualization, P.Z. and L.L.; Supervision, R.-Y.Y.; Project administration, R.-Y.Y.; Funding acquisition, D.C. and M.-H.W. All authors have read and agreed to the published version of the manuscript.

**Funding:** This work was supported by the Putian University's Initiation Fee Project for Importing Talents for Scientific Research (2019001) and (2019003).

**Acknowledgments:** The authors acknowledge Chin-Yi Tsai, Hong-Zheng Lai and Shih-Kun Liu for the help with sample measurement.

**Conflicts of Interest:** The authors declare no conflicts of interest.

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
