# Peer review of "Application of a Stub-Loaded Square Ring Resonator for Wideband Bandpass Filter Design"

_electronics, doi:10.3390/electronics9010176_

Round 1
Reviewer 1 Report
This study investigates wideband bandpass filter which consists a square ring loaded resonator with two stubs. The wideband filter is manufactured and results are shown which has a good agreement with simulation results.
Authors answered the most of the comments but in order for manuscript to be more readable, please use the following structure.
Please organize the introduction as follows. Then explicitly provide more details on the contribution part regarding how the proposed method is better than the current state of the art literature.
I. Introduction, I.A. Related Work, I.B. Contributions, I.C. Organization of the Paper.
Author Response
The comment is well received. In this revised version, we have organized the introduction as the suggestions by the reviewer and described more details on the contribution part to show how the proposed method is better than the current state of the art literature, as the mark parts with yellow color. Moreover, a I.C. section for the organization of the paper have been added in the revised version to be more readable.
In addition, English have been checked carefully again to reduce the typing or spelling errors.
Reviewer 2 Report
The revised version of the manuscript has been improved and my concerns clarified.
I have no suggestions.
Author Response
Thanks for the reviewer.
English have been checked carefully again to reduce the typing or spelling errors.
Round 2
Reviewer 1 Report
This work can be accepted in its current form.
This manuscript is a resubmission of an earlier submission. The following is a list of the peer review reports and author responses from that submission.
Round 1
Reviewer 1 Report
The proposed band pass filter base on a square ring resonator is interesting but non innovative. The innovations introduced in this work should be better specified in the introduction section because in this form they are not enought clear.
Moreover the following suggestions aimed at improve the work quality should be take into account by the authors.
1) the obtained results are non so satisfactory. From the comparative table reported in the manucript the obtained results in terms of bandwidth and return are not so good. Please comment.
2) the data reported in figure 9b are not so readable. You should use different colours for the return and insertion loss data.
3) please introduce further description of the experimental setup.
4) the reference section turn out to be too much limited and it should be strongly improved.
Reviewer 2 Report
This study investigates wideband bandpass filter which consists a square ring loaded resonator with two stubs. The wideband filter is manufactured and results are shown which has a good agreement with simulation results.
Authors have a recent paper “An Ultra-Wideband Bandpass Filter with a Notch Band and Wide Upper Bandstop Performances”, please detail more of your differences and similarities with this work.
Please organize the introduction as follows. Then explicitly provide more details on the contribution part regarding how the proposed method is better than the current state of the art literature including the aforementioned work.
I. Introduction, I.A. Related Work, I.B. Contributions, I.C. Organization of the Paper.